# Exploring the Effect of Arsenic-Containing Hydrocarbon on the Bidirectional Synaptic Plasticity of the Dorsal Hippocampus

**DOI:** 10.3390/ijms25137223

**Published:** 2024-06-29

**Authors:** Chunxiao Tian, Yenan Qi, Yu Zheng, Pei Xia, Qiwen Liu, Mengying Luan, Junyao Zheng, Rujuan Song, Meng Wang, Dejiao Qi, Chan Xiong, Lei Dong

**Affiliations:** 1School of Life Sciences, Tiangong University, Tianjin 300387, China; tianxiaochunun@163.com (C.T.); 2230070915@tiangong.edu.cn (Y.Q.); zhengyu@tiangong.edu.cn (Y.Z.); 2231101349@tiangong.edu.cn (M.L.); 2231101352@tiangong.edu.cn (J.Z.); 2230101314@tiangong.edu.cn (R.S.); 2231101339@tiangong.edu.cn (M.W.); 2231101325@tiangong.edu.cn (D.Q.); 2School of Biomedical Engineering, Tianjin Medical University, Tianjin 300070, China; 3School of Electronics & Information Engineering, Tiangong University, Tianjin 300387, China; 4School of Control Science and Engineering, Tiangong University, Tianjin 300387, China; 2131121257@tiangong.edu.cn; 5School of Brain Science and Brain Medicine, Zhejiang University, Hangzhou 310012, China; 12318278@zju.edu.cn; 6Analytical Chemistry, Institute of Chemistry, University of Graz, 8010 Graz, Austria; 7BOKU Core Facility Mass Spectrometry, University of Natural Resources and Life Sciences (BOKU), 1190 Vienna, Austria

**Keywords:** Arsenic-containing hydrocarbons, dorsal hippocampus, bidirectional synaptic plasticity, long-term potentiation (LTP), long-term depression (LTD)

## Abstract

Arsenic-containing hydrocarbons (AsHCs) are common in marine organisms. However, there is little research on their effects on the central nervous system’s advanced activities, such as cognition. Bidirectional synaptic plasticity dynamically regulates cognition through the balance of long-term potentiation (LTP) and long-term depression (LTD). However, the effects of AsHCs on bidirectional synaptic plasticity and the underlying molecular mechanisms remain unexplored. This study provides the first evidence that 15 μg As L^−1^ AsHC 360 enhances bidirectional synaptic plasticity, occurring during the maintenance phase rather than the baseline phase. Further calcium gradient experiments hypothesize that AsHC 360 may enhance bidirectional synaptic plasticity by affecting calcium ion levels. The enhancement of bidirectional synaptic plasticity by 15 μg As L^−1^ AsHC 360 holds significant implications in improving cognitive function, treating neuro-psychiatric disorders, promoting neural recovery, and enhancing brain adaptability.

## 1. Introduction

Arsenic (As) is a widely distributed semi-metallic chemical element in nature, existing in both inorganic and organic forms. Inorganic arsenic (iAs), recognized as a human carcinogen, mainly enters the human body through drinking water and diet. Extended exposure to iAs can cause significant harm to human organs and the nervous system, leading to cancer, cognitive or intellectual impairments [1], and impaired spatial learning and memory abilities [2,3]. Organic arsenic typically includes water-soluble arsenic and lipid-soluble arsenicals (arsenolipids), with dietary seafood being the primary route of human exposure to organic arsenic. Compared to iAs, water-soluble arsenicals such as arsenobetaine and arsenosugars have no toxic effects on humans [4]. Arsenic-containing hydrocarbons (AsHCs) are a type of arsenolipid, varying by specific compounds. AsHCs possess high biological activity and are commonly found in fish and algae [5,6,7]. AsHC 360 is a specific compound of AsHCs. Meyer et al. studied the effects of AsHC 360 on human cells through in vitro culture of bladder and liver cells [6]. In vivo experiments on fruit flies demonstrated that AsHCs affect the late developmental stages of fruit flies and accumulate in the brain [8,9]. Furthermore, studies using a blood-brain barrier model have shown that AsHC 360 increases the permeability of the blood-brain barrier [10]. Tracking and detection have revealed the migration of AsHC 360 and other arsenolipids from seafood to breast milk [11,12]. This suggests that AsHC 360 and other arsenic compounds can enter infants’ bodies through breast milk, potentially affecting brain development. However, the same effect was not observed in milk from cows fed with seaweed [13]. During critical early developmental stages, the infant brain may be sensitive to external arsenolipid compounds. Therefore, this study investigates the influence of AsHC 360 on cognition, starting from higher neural activities associated with cognition.

The deep hippocampal region of the brain serves as a critical nexus for advanced neural activities, including cognition processes. Memory loss phenomena have been observed following hippocampal damage, which underscores the critical role of the hippocampus in memory function [14]. Within the hippocampus, the dorsal hippocampus (DH) plays a crucial role in spatial memory, episodic memory, learning, and memory integration, exerting a significant influence on overall cognitive function [15,16,17]. The hippocampus comprises a classical tri-synaptic neural pathway Perforant Path (PP)-Dentate Gyrus (DG)-CA3-CA1 [18]; among these regions, the Schaffer-CA1 synapse plays a pivotal role in the final processing and synthesis of information across the entire neural pathway [19]. The bidirectional synaptic plasticity of the Schaffer-CA1 synapse is an important model for studying cognition. Bidirectional synaptic plasticity refers to the ability of synaptic connections to change in strength in two opposite directions and is the fundamental mechanism underlying neural adaptation and learning [20]. Bidirectional synaptic plasticity primarily manifests in two forms: long-term potentiation (LTP) and long-term depression (LTD) [21]. LTP involves the sustained enhancement of synaptic transmission, while LTD involves the opposite, the suppression of synaptic transmission. Both LTP and LTD can be further divided into induction and maintenance phases. The induction phase refers to the moment when stimulation is applied to Schaffer collaterals, which causes synapses to transition from a resting state to a functional state. The maintenance phase refers to the later stable enhancement and weakening of LTP and LTD following stimulation. Each phase has distinct mechanisms and requirements, contributing to their unique roles in bidirectional synaptic plasticity [22,23,24]. The molecular mechanisms involved in this process have significant implications for cognitive functions such as learning and memory. Calcium ions (Ca^2+^) play a crucial role in regulating both LTP and LTD, acting as a key mediator linking neuronal activities to changes in synaptic plasticity [25,26]. The induction of both LTP and LTD relies on changes in intracellular Ca^2+^ levels in postsynaptic neurons [27]. Activation of NMDA receptors, which are calcium-permeable ion channels, leads to the influx of Ca^2+^ into postsynaptic neurons. This influx of Ca^2+^ is a critical step in the induction of both forms of synaptic plasticity. The magnitude and duration of Ca^2+^ influx play a crucial role in determining the direction of synaptic plasticity, whether it involves enhancement (LTP) or suppression (LTD), as well as the subsequent maintenance phase. Although the mechanisms of calcium ion action differ in LTP and LTD, its central role in regulating synaptic strength remains unchanged. The entry of calcium ions and activation of downstream signaling pathways are critical steps in synaptic plasticity. By modulating calcium influx and its signaling pathways, neurons can dynamically adjust synaptic connections, thereby playing a crucial role in learning and memory processes. Therefore, calcium homeostasis plays a fundamental role in shaping the plasticity of the nervous system and underlies processes such as cognition.

In this study, we applied 15 μg As L^−1^ AsHC 360 to dorsal hippocampal slices from Sprague-Dawley rats and recorded field excitatory postsynaptic potentials (fEPSPs) in the CA1 region of the hippocampus using a multi-electrode array system (MEA), thereby extending previous research on synaptic plasticity [28]. Specifically, we administered a concentration of 15 μg As L^−1^ AsHC 360 to assess its effects on bidirectional synaptic plasticity. Furthermore, we conducted experiments to explore the involvement of Ca^2+^ activity in the bidirectional synaptic plasticity changes induced by AsHC 360. By investigating the interplay between AsHC 360 exposure and Ca^2+^ dynamics within the context of bidirectional synaptic plasticity, we aim to gain a deeper understanding of the underlying mechanisms through which AsHC 360 may impact neural function. This study can provide valuable insights into the effects of AsHC 360 on the central nervous system and cognitive processes such as learning and memory.

## 2. Results

### 2.1. AsHC 360 Enhances HFS-LTP

The experimental setup is illustrated in Figure 1. Rat hippocampal slices were placed in the well of a glass electrode chamber (Figure 1F) and positioned on the MEA electrophysiological recording platform (Figure 1A). The MEA electrophysiological recording platform was controlled by computer software to apply high-frequency stimulation (HFS) to the Schaffer-CA1 neural pathway to induce HFS-LTP. The LTP-Director software was used to record the field excitatory postsynaptic potentials (fEPSPs) in the CA1 region of the hippocampus. The experimental protocol is illustrated in Figure 2A. In the control group (without AsHC 360), baseline fEPSPs were recorded for 20 min, followed by recording LTP-fEPSPs for 100 min (total time 120 min). In the post AsHC 360 group, AsHC 360 was introduced via a perfusion apparatus at t = 30 min after the stabilization of the LTP-fEPSPs baseline (baseline recorded for 20 min, followed by 100 min of recording after HFS-LTP induction, total time 120 min). In the pre-AsHC 360 group, AsHC 360 was introduced via a perfusion apparatus at t = −10 min before LTP induction (baseline recorded for 20 min, followed by 100 min of recording after HFS-LTP induction, total time of 120 min).

The impact of post AsHC 360 on HFS-LTP is depicted in Figure 2B (red circle/bars). A solution containing AsHC 360 was introduced via perfusion apparatus at t = 30 min, subsequent to the stabilization of LTP. During the initial 3 min following introduction, no significant alteration in fEPSP value was observed. Subsequently, fEPSP values exhibited a steady increase, culminating in a 0.2 increase within 10 min. By 43 min, fEPSP values had reached a stable plateau. We found that AsHC 360 enhanced the fEPSPs of HFS-LTP.

Whisker plots were presented as depicted in Figure 2C. In stark contrast to controls, Con-1 and Post-1 were compared to the amplitude from the corresponding time period for the control recording. Slices that had not been exposed to AsHC 360 showed no significant difference between recordings (0–30 min) (Post-1 vs. Con-1). However, distinctions were observed between the control and post AsHC 360 (40–100 min) (Post-2 vs. Con-2; *p* < 0.001), indicating a significant increase following the introduction of AsHC 360 compared to the control group.

The impact of pre AsHC 360 on HFS-LTP is illustrated in Figure 2B (blue triangle/bars). A solution of AsHC 360 was introduced via perfusion apparatus at t = −10 min, preceding the induction of LTP. Following the introduction, there was no discernible change in fEPSPs (−10 min to 0 min). LTP was subsequently induced at t = 0 min. In comparison to the control experimental group, the fEPSPs amplitude remained consistent for 7 min (from 3 to 10 min). Subsequently, after 10 min, LTP-fEPSPs gradually increased, reaching a 0.2 increase within 8 min. By the 18 min, fEPSPs remained stable. Notably, AsHC 360 was observed to enhance the fEPSP values of HFS-LTP. Despite the pre-introduction of AsHC 360 at t = −10 min before HFS-LTP induction, there was no significant alteration in fEPSPs.

Whisker plots were depicted as presented in Figure 2C. In stark contrast to controls, Con-1 and Pre-1 were juxtaposed with the amplitude from the corresponding time period for the control recording. Despite exposure to AsHC 360, no significant difference was observed in LTP-fEPSPs of brain slices (−10–0 min) (0–10 min; Pre-1 vs. Con-1). Interestingly, starting from t = 18 min, disparities emerged between the control and pre AsHC 360 (20–100 min) (Pre-2 vs. Con-2), indicating a significant increase subsequent to the addition of AsHC 360 compared to the control group.

### 2.2. AsHC 360 Enhances TBS-LTP

The introduction protocol is illustrated in Figure 3A. TBS was administered to the Schaffer collateral to induce TBS-LTP (Figure 1E), with fEPSPs recorded in CA1. Additional control experiments were conducted without any AsHC 360 (black square/bars).

The impact of post AsHC 360 on TBS-LTP is depicted in Figure 3B (red circle/bars). A solution of AsHC 360 was introduced via perfusion apparatus at t = 30 min, subsequent to the stabilization of LTP. Within the initial 13 min post-introduction, the fEPSPs amplitude exhibited a steady increase, reaching a rise of 0.21. Following this, fEPSP values remained stable after 43 min. Whisker plots were displayed as shown in Figure 3C. In notable contrast to controls, Con-2 and Post-2 were juxtaposed with the amplitude from the corresponding time period for the control recording. Slices subjected to AsHC 360 showed displayed a significant increase in LTP-fEPSPs (40–100 min, Con-2 vs. Post-2, *p* < 0.001).

The impact of pre AsHC 360 on TBS-LTP is illustrated in Figure 3B (blue triangle/bars). A solution of AsHC 360 was introduced via perfusion apparatus at t = −10 min, preceding LTP induction. Following induction, the fEPSPs values of TBS-LTP remained stable for 8 min (10–18 min). Subsequently, at the 18-min mark, there was a stable increase, rising by 0.21 (from 18 to 26 min). Following this, fEPSPs remained stable after 26 min. In marked contrast to controls, Con-2 and Pre-2 were juxtaposed with the amplitude from the corresponding time period for the control recording. Slices subjected to AsHC 360 displayed a significant increase in LTP-fEPSPs values (40–100 min, Con-2 vs. Pre-2, *p* < 0.001). Con-1 and Pre-1 were compared to the amplitude from the same time point of the control recording. Slices that had experienced AsHC 360 showed no significant difference between recordings (−10–0 min) (0–10 min, Pre-1 vs. Con-1). This finding aligns with our previous experiment results depicted in Figure 2A under HFS-LTP.

### 2.3. AsHC 360 Reduces LFS-LTD

In this section, we investigate the potential impact of AsHC 360 on long-term depression (LTD) induced by low-frequency electrical stimulation (LFS). To address this, we devised experimental protocol three, as illustrated in Figure 4A. The total recording time for LTD-fEPSPs spas is 130 min.

In the control group experiment (without AsHC 360), baseline fEPSPs were recorded for 30 min, followed by the recording of LTD-fEPSPs for an additional 100 min.

The impact of post AsHC 360 on LTD is depicted in Figure 4B (red square/bars). A solution of AsHC 360 was introduced via perfusion apparatus at t = 30 min after stabilization of LTD. Within the initial 8 min post-introduction, the fEPSPs value decreased by 0.24. Subsequently, from t = 40 min onwards, LTD-fEPSPs remained stable. Our observation indicates that AsHC 360 enhances LFS-LTD, resulting in lower LTD-fEPSPs.

The statistical analysis plot is depicted in Figure 4C. We compared the amplitudes of Con-1 and Post-1 during the same time period and found no significant difference in recorded amplitudes for slices that had not experienced AsHC 360 (Post-1 vs. Con-1, *p* > 0.05) (10 min–30 min). However, differences emerged 10 min after the introduction of AsHC 360 (t = 40 min). Significant differences were observed in recorded amplitudes between the control group and the post AsHC 360 group (Post-2 vs. Con-2, *p* < 0.001) (40 min–100 min).

The impact of pre AsHC 360 on LTD is illustrated in Figure 4B (blue circle/bars). AsHC 360 was introduced at t = −20 min before inducing LTD, which was 10 min earlier compared to the pre AsHC 360 group shown in Figure 3. This adjustment was made to accommodate the longer duration required for LTD induction (LFS: Duration: 10 min, Pulse Freq: 1 Hz.) compared to LTP induction (HFS: 100 pulses delivered at 100 Hz for 1 s). LTD induction is initiated at t = −10 min. Following the induction of LTD, the amplitude of LTD-fEPSPs remained stable for 7 min (3–10 min). At t = 10 min, the amplitude of LTD-fEPSPs gradually decreased, reaching a decrease of 0.21 within 7 min (10–17 min). After t = 20 min, the LTD-fEPSPs remained stable. We observed that AsHC 360 facilitated the LFS-LTD. The prior introduction of AsHC 360 at t = −20 min did not result in significant changes in fEPSPs before inducing LFS-LTD.

Whisker plots were presented as depicted in Figure 3C. In stark contrast to the control group, Con-1 and Pre-1 were evaluated against amplitudes recorded during the same time period of the control recording. Despite experiencing AsHC 360, there were no significant differences observed in LTD-fEPSPs of brain slices (−10–0 min) (0–10 min; Pre-1 vs. Con-1; *p* > 0.05). Interestingly, after t = 18 min, differences emerged between the control and pre AsHC 360 (20–100 min) (Pre-2 vs. Con-2), indicating a significant reduction after the introduction of AsHC 360 in comparison to the control group.

### 2.4. Ca^2+^ Activity Was Involved in the Process of LTP Enhancement Caused by Post AsHC 360

The experimental procedures, delineated in Figure 5A,D, involved the following aspects. 

As shown in the experimental protocol ① in Figure 5A, which corresponds to the experimental results EG1 (black square/bars) in Figure 5B, the results demonstrated that during the induction and recording of LTP in ACSF with a low concentration of 2 mM [Ca^2+^]_e_ (−20–30 min), the LTP-fEPSPs approximated 1.37. Subsequent transition to ACSF with a normal concentration of 2.5 mM [Ca^2+^]_e_ (t = 30 min) led to a 0.13 increment in LTP, which then stabilized. At t = 60 min, the introduction of AsHC 360 prompted a further increase of 0.2 in LTP-fEPSPs, which was sustained thereafter.

As shown in the experimental protocol ② in Figure 5A, which corresponds to the experimental results EG2 (red circle/bars) in Figure 5B, it was demonstrated that during the induction and recording of LTP in ACSF with a normal concentration of 2.5 mM [Ca^2+^]_e_ (−20–30 min), the LTP-fEPSPs was around 1.52. Transitioning to ACSF with a high concentration of 3 mM [Ca^2+^]_e_ (t = 30 min) resulted in a decrease of 0.14 in LTP, which subsequently stabilized. At t = 60 min, the introduction of AsHC 360 induced a two-stage decrease in LTP-fEPSPs. During this interval, LTP decreased by 0.38 and sustained after t = 85 min.

Statistical analysis of LTP-fEPSPs is shown in Figure 5C. The results of one-way ANOVA were as follows: EG1-1 vs. EG1-2, *p* < 0.001; EG1-2 vs. EG1-3, *p* < 0.001; EG2-1 vs. EG2-2, *p* < 0.001; EG2-2 vs. EG2-3, *p* < 0.001. These results confirm significant differences in LTP-fEPSPs among ACSF solutions with varying Ca^2+^ concentrations and in the presence of AsHC 360.

The analysis of this experimental group unveiled that an increase in extracellular Ca^2+^ concentration promotes enhancement in LTP-fEPSPs under reasonable extracellular Ca^2+^ concentrations. Conversely, excessively high Ca^2+^ concentrations were observed to inhibit LTP occurrence. Notably, ACSF with 2.5 mM [Ca^2+^]_e_ emerged as the optimal concentration for inducing LTP among the varied Ca^2+^ concentrations examined. Moreover, under reasonable extracellular Ca^2+^ concentrations, the addition of AsHC 360 elicited a further increase in LTP-fEPSPs, indicating its promotive effect on LTP excitatory postsynaptic potentials. Conversely, under a high concentration of 3 mM [Ca^2+^]_e_ ACSF, AsHC 360 was found to inhibit the enhancement of LTP excitatory postsynaptic potentials.

As shown in the experimental protocol ① in Figure 5D, which corresponds to the experimental results EG3 (black square/bars) in Figure 5E, it was indicated that during the induction and recording of LTP in ACSF with a low concentration of 2 mM [Ca^2+^]e (−20–30 min), the LTP-fEPSPs approximated 1.37. Following the addition of AsHC 360 (t = 30 min), LTP increased by 0.12 and remained stable. At t = 60min, with the addition of a normal concentration of 2.5 mM [Ca^2+^]_e_ ACSF (60–100min), LTP-fEPSPs increased by 0.21 and remained stable. Comparatively, the transition from low concentration to normal concentration of cerebrospinal fluid resulted in a 0.13 increase in fEPSPs (as observed in Figure 5A-①). In contrast, with the addition of AsHC 360 in Figure 5D-EG3, the upward trend of fEPSPs was enhanced, increasing by 0.21, which was an increase of 0.08 over the original foundation.

As shown in the experimental protocol ② in Figure 5D, which corresponds to the experimental results EG4 (red circle/bars) in Figure 5E, it was indicated that during the induction and recording of LTP in ACSF with a normal concentration of 2.5 mM [Ca^2+^]_e_ (−20–30 min), the LTP-fEPSPs approximated 1.52. Subsequently, the addition of AsHC 360 (t = 30 min) led to an increase of 0.19 in LTP, which remained stable. At t = 60 min, transitioning to a high concentration of 3 mM [Ca^2+^]_e_ ACSF (60–100 min) resulted in a decrease of 0.51 in LTP-fEPSPs, which stabilized after t = 80 min. Compared to the transition observed in Figure 5A-②, wherein the transition from normal concentration to high concentration cerebrospinal fluid resulted in a 0.14 decrease in fEPSPs, the addition of AsHC 360 shown in Figure 5D-EG4, enhanced the downward trend of fEPSPs, decreasing by 0.37 compared to the original foundation.

Statistical analysis of LTP-fEPSPs is presented in Figure 5F. The results of one-way ANOVA were as follows: EG3-1 vs. EG3-2, *p* < 0.001; EG3-2 vs. EG3-3, *p* < 0.001; EG4-1 vs. EG4-2, *p* < 0.001; EG4-2 vs. EG4-3, *p* < 0.001. These results affirm significant differences in LTP-fEPSPs among varied Ca^2+^ concentrations of ACSF and the presence of AsHC 360.

Comparison between experimental groups EG-1 and EG-3 revealed that during the transition from low Ca^2+^ concentration to normal Ca^2+^ concentration, the addition of AsHC 360 resulted in a more significant increase in LTP-fEPSPs compared to when AsHC 360 was absent. Similarly, a comparison between experimental groups EG-2 and EG-4 demonstrated that during the transition from normal Ca^2+^ concentration to high Ca^2+^ concentration, the addition of AsHC 360 led to a more pronounced decrease in LTP-fEPSPs compared to when AsHC 360 was absent.

### 2.5. Ca^2+^ Activity Was Involved in the Process of LTD Enhancement Caused by Post AsHC 360

The experimental procedures, delineated in Figure 6A,D, involved the following aspects. 

As shown in the experimental protocol ① in Figure 6A, which corresponds to the experimental results EG5 (black square/bars) in Figure 6B, it was demonstrated that during the induction and recording of LTD in ACSF with a low concentration of 2 mM [Ca^2+^]_e_ (−30–30 min), the LTD-fEPSPs approximated 0.65. Subsequent transition to ACSF with a normal concentration of 2.5 mM [Ca^2+^]_e_ (t = 30 min) led to a 0.13 decrement in LTD-fEPSPs, which then stabilized. At t = 60 min, the introduction of AsHC 360 (60–100 min) prompted a further decrease of 0.20 in LTD-fEPSPs and stabilized from t = 70 min. 

The experimental protocol ② shown in Figure 6A corresponds to the experimental results EG6 (red circles/bars) in Figure 6B. These results demonstrated that during the induction and recording of LTD in ACSF with a normal concentration of 2.5 mM [Ca^2+^]_e_ (−30–30 min), the LTD-fEPSPs value approximated 0.55. Subsequent transition to ACSF with a high concentration of 3 mM [Ca^2+^]_e_ ACSF (t = 30 min) led to a 0.11 increment in LTD, which then stabilized. At t = 60 min, the introduction of AsHC 360 (60–100 min) induced a two-stage increase in LTD-fEPSPs. During this interval, LTD increased by 0.23 and sustained after t = 82 min.

According to the statistical analysis in Figure 6C, EG5-1 vs. EG5-2, *p* < 0.001; EG5-2 vs. EG5-3, *p* < 0.001; EG6-1 vs. EG6-2, *p* < 0.001; EG6-2 vs. EG6-3, *p* < 0.001. These results confirm significant differences in LTD-fEPSPs among ACSF solutions with varying Ca^2+^ concentrations and in the presence of AsHC 360.

In this experimental group, we observed that within a reasonable extracellular Ca^2+^ concentration, an increase in [Ca^2+^]_e_ concentration promoted the occurrence of LTD, resulting in a decrease in LTD-fEPSPs. However, excessively high Ca^2+^ concentrations inhibited the occurrence of LTD. Among the experimental groups with varied Ca^2+^ concentrations of artificial cerebrospinal fluid, 2.5 mM [Ca^2+^]_e_ emerged as the optimal concentration for inducing LTD. Under conditions of reasonable extracellular Ca^2+^ concentration, associated with LTD induction characterized by a decrease in excitatory postsynaptic potentials, we observed a further decrease in LTD-fEPSPs following AsHC 360 infusion, indicating that AsHC 360 promotes the decline of LTD excitatory postsynaptic potentials. Conversely, under high 3 mM [Ca^2+^]_e_ ACSF conditions, associated with LTD induction characterized by an increase in excitatory postsynaptic potentials, we found a further increase in LTD-fEPSPs after AsHC 360 infusion, suggesting that AsHC 360 inhibits LTD.

The experimental protocol ① shown in Figure 6D corresponds to the experimental results EG7 (black squares/bars) in Figure 6E. During the induction and recording of LTD (−30–30 min) with a low concentration of 2 mM [Ca^2+^]_e_ ACSF, the LTD-fEPSPs approximated 0.64. The subsequent introduction of AsHC 360 (t = 30 min) led to a decrease of 0.15 in LTD-fEPSPs, which then remained stable. Following the transition to a normal concentration of 2.5 mM [Ca^2+^]_e_ ACSF (60–100 min) resulted in a decrease of 0.21 in LTD-fEPSPs, which was then sustained. In comparison with the experimental protocol in Figure 6A-① (Figure 6B, black dataset, EG-5), where the transition from low concentration to normal concentration cerebrospinal fluid resulted in a decrease of 0.13 in LTD-fEPSPs. In contrast, in Figure 6E-EG7, after the addition of AsHC 360, the downward trend of LTD-fEPSPs values intensified, with a decrease of 0.21, representing an increase of 0.08 in the level of decline compared to the baseline.

The experimental protocol ② shown in Figure 6D corresponds to the experimental results EG8 in Figure 6E. During the induction and recording of LTD (−30–30 min) with a normal concentration of 2.5 mM [Ca^2+^]e ACSF, the LTD-fEPSPs approximated 0.55. Subsequent to the introduction of AsHC 360 (t = 30 min), LTD decreased by 0.21 and remained stable thereafter. Following the transition to a high concentration of 3 mM [Ca^2+^]_e_ ACSF (60–100 min), an increase of 0.14 in LTD-fEPSPs was observed, which remained stable after t = 67 min. In comparison with the experimental protocol in Figure 6A-② (Figure 6B, red dataset, EG-6), where the transition from normal concentration to high concentration cerebrospinal fluid resulted in an increase of 0.11 in LTD-fEPSPs. In contrast, in Figure 6E-EG8, after the addition of AsHC 360, the upward trend of LTD-fEPSPs intensified, with an increase of 0.14, representing an increase of 0.03 in the level of ascent compared to the baseline.

The statistical analysis graph of LTD-fEPSPs is shown in Figure 6F. The results of one-way ANOVA were as follows: EG7-1 vs. EG7-2, *p* < 0.001; EG7-2 vs. EG7-3, *p* < 0.001; EG8-1 vs. EG8-2, *p* < 0.001; EG8-2 vs. EG8-3, *p* < 0.001. These results affirm significant differences in LTD-fEPSPs among varied ACSF calcium concentrations and AsHC 360 treatments.

Through the comparison of experimental groups EG-5 and EG-7, we observed that during the transition from low Ca^2+^ concentration to normal Ca^2+^ concentration, the addition of AsHC 360 resulted in a more significant decrease in LTD-fEPSPs compared to when AsHC 360 was absent. Similarly, through the comparison of experimental groups EG-6 and EG-8, we found that during the transition from normal Ca^2+^ concentration to high Ca^2+^ concentration, the addition of AsHC 360 led to a more pronounced increase in LTD-fEPSPs compared to when AsHC 360 was absent.

## 3. Discussion

In this study, we employed Sprague-Dawley rat brain slices as our experimental model and utilized a multi-electrode array (MEA) system to record neurophysiological signals. We investigated the effects of 15 μg As L^−1^ AsHC 360 on bidirectional synaptic plasticity. Initially, the influence of AsHC 360 on LTP was examined under various induction protocols, revealing that its impact occurred predominantly during the maintenance phase rather than the baseline stage. Subsequently, the exploration of AsHC 360’s effect on LTD indicates that its action period occurs during the maintenance phase rather than the baseline stage. Finally, through experiments manipulating Ca^2+^ concentration, we hypothesize that AsHC 360 may enhance bidirectional synaptic plasticity by influencing calcium ion levels.

### 3.1. AsHC 360 Facilitates Bidirectional Synaptic Plasticity during the Maintenance Phase Rather Than the Baseline Stage

The electrophysiological results depicted in Figure 1B, Figure 2B, and Figure 3B indicate that irrespective of the induction method (HFS for LTP, TBS for LTP, or LFS for LTD), the influence of AsHC 360 on bidirectional synaptic plasticity manifests predominantly during the maintenance phase rather than the baseline segment. The distinctions between the baseline and maintenance phases are evident. During the baseline stage, even with the prior introduction of AsHC 360, the absence of high-frequency stimulation induction prevents silent synapses from transitioning into functional ones, resulting in negligible synaptic depolarization, with Magnesium ions (Mg^2+^) blocking NMDA receptor channels, preventing the transmission of complex signals within the synapse and thereby failing to affect synaptic plasticity [29,30]. 

Compared to the baseline stage, during the maintenance phase of bidirectional synaptic plasticity, synapses are in a functional state, maintaining persistent enhancement or weakening. As an arsenolipid, AsHC 360 may provide favorable conditions for membrane fusion in cognitive processes [6,10], increasing synaptic transmission and enhancing the maintenance phase of bidirectional synaptic plasticity [28]. This influence may involve multiple molecular mechanisms, such as modulation of neurotransmitter receptors, alterations in intracellular signaling pathways, and modifications of postsynaptic membrane proteins, among others. These changes likely facilitate long-term synaptic modulation, consequently affecting cognitive information transmission and processing in the brain.

### 3.2. Different [Ca^2+^]_e_ Has Different Effects on Bidirectional Synaptic Plasticity

Calcium ions play a crucial role in the maintenance phase of bidirectional synaptic plasticity by influencing the activity of protein kinases such as CaMKII and Calcineurin, as well as regulating the insertion and internalization of AMPARs, thereby maintaining the synaptic enhancement or weakening state. 

From the results of the Ca^2+^ experiment, successful induction of both LTP and LTD in artificial cerebrospinal fluid (ACSF) with a low concentration was observed. Switching the perfusion to normal concentration ACSF to increase the Ca^2+^ concentration in the environment promoted bidirectional synaptic plasticity. From an ionic mechanism perspective, during the maintenance phase of LTP, the increase in Ca^2+^ concentration facilitated the activation of calcium/calmodulin-dependent protein kinase II (CaMKII) [31]. CaMKII phosphorylates AMPA receptors, increasing their activity or promoting their translocation to the postsynaptic membrane, thereby enhancing LTP synaptic transmission efficiency. In the maintenance phase of LTD, the increase in Ca^2+^ concentration promoted the activation of calcineurin, which dephosphorylates specific proteins such as AMPA receptors, causing their removal from the postsynaptic membrane, thereby enhancing LTD synaptic transmission [32,33]. When LTP and LTD were successfully induced in ACSF with normal concentration, switching to high-concentration ACSF using perfusion resulted in a slight decrease in the maintenance phase of LTP and LTD. This indicates that the Ca^2+^ concentration has reached excessive levels, inhibiting bidirectional synaptic plasticity. The high Ca^2+^ concentration reduces the transmission efficiency of bidirectional synaptic plasticity. The intracellular Ca^2+^ concentration is already at a supersaturated level.

### 3.3. AsHC 360 Promotes Bidirectional Synaptic Plasticity by Influencing Ca^2+^ Levels

In this section, We investigated the potential impact of AsHC 360 on Ca^2+^ during the maintenance phase of bidirectional synaptic plasticity. In the gradient experiment comparing different Ca^2+^ concentrations following the addition of AsHC 360, a significant enhancement in LTP and LTD was observed compared to those without the AsHC 360 group. Based on this phenomenon, we hypothesize that the compound AsHC 360 may enhance bidirectional synaptic plasticity by affecting calcium ion levels. This observation, combined with previous findings by Meyer et al. suggesting AsHCs’ interference with cell membrane integrity [6], and Müller et al.’s indicating of significant local morphological abnormalities and gap formation between cells following AsHC treatment [10], suggests a higher likelihood of AsHC 360 binding to neuronal synaptic cell membrane with increased affinity and bioavailability. Therefore, AsHC 360 may enhance cell membrane permeability (fluidity), thereby expanding the ion channels between intracellular and extracellular environments. This phenomenon might promote Ca^2+^ influx, with calcium ions mediating the enhancement of bidirectional synaptic plasticity by AsHC 360. The sustained Ca^2+^ influx activates LTP and LTD protein kinases, thereby increasing synaptic transmission efficiency [29,34]. Thus, this strengthens the maintenance phase of bidirectional synaptic plasticity.

In the presence of excessively high extracellular Ca^2+^ concentration stabilized at 3mM Ca^2+^, this study observed a progressive decline in LTP following the addition of AsHC 360, characterized by two successive decreases ultimately approaching baseline levels. It is hypothesized that 15 μg As L^−1^ AsHC 360 increases Ca^2+^ influx, by increasing Ca^2+^ influx, perpetuates this state of excessive Ca^2+^ concentration, thereby inhibiting the maintenance period of LTP. The disruption of Ca^2+^ homeostasis may render ex vivo hippocampal slices intolerant to further Ca^2+^ influx, resulting in fluctuations in the maintenance period of LTP and eventual near return to baseline, indicative of impending loss of synaptic information transmission function, i.e., LTP disappearance. On the other hand, in the presence of normal extracellular Ca^2+^ concentration stabilized at 2.5 mM Ca^2+^, a sustained increase in LTP was observed following the addition of 15 μg As L^−1^ AsHC 360. Since 15 μg As L^−1^ AsHC 360 may enhance Ca^2+^ influx, continued promotion of Ca^2+^ influx following its addition further augments the maintenance period of LTP until reaching a new equilibrium. Similarly, the observed effects on LTD are consistent with those on LTP. 15 μg As L^−1^ AsHC 360 promotes both LTD and LTP, thereby enhancing their transmission efficiency bidirectionally, such that promotion further enhances and inhibition further suppresses. 

The enhancement of bidirectional synaptic plasticity by 15 μg As L^−1^ AsHC 360 is of significant importance. In the future, this could provide valuable insights into improving cognitive function, treating neuropsychiatric disorders, promoting neural recovery, and enhancing brain adaptability. By understanding the mechanisms through which AsHC 360 enhances bidirectional synaptic plasticity, we can explore new therapies leveraging the potential of this compound. This paves the way for the development of innovative treatments that can effectively regulate bidirectional synaptic plasticity, ultimately leading to better outcomes in cognitive enhancement and neural health.

## 4. Materials and Methods

### 4.1. Animals

This study utilized male Sprague-Dawley (SD) rats aged 15–18 days and weighing 25–30 grams, purchased from Beijing Vital River Laboratory Animal Technology Co., Ltd. The license number for their use is SYXK (Jin)-2019–0002. The experimental rats were delivered by the company’s truck, which departed at 8 AM and arrived at 12 PM. The truck was equipped with a temperature control system to maintain an appropriate temperature range (typically 20–26 °C) to prevent the animals from overheating or becoming too cold. The truck also had a good ventilation system. The transport cages were securely fixed in the truck to prevent movement and tipping during transport, thereby minimizing potential stress to the animals. Experiments were conducted on the second day after the rats’ arrival to allow them to acclimate to the new environment and alleviate stress effects. The experimental animals were housed in a clean and tidy animal room, maintained under a 24-h light-dark cycle, with the temperature controlled at 23–25 °C and relative humidity maintained at 40–60%. The rats lived in individual cages with free access to water and food. The experiments were completed within three days of the animals’ arrival. All experimental procedures and ethical standards in this study were approved by the Experimental Animal Ethics Committee of the Institute of Radiation Medicine, Chinese Academy of Medical Sciences (Tianjin, China), ensuring the rationality and ethical compliance of the experiments.

### 4.2. Experimental Pharmaceuticals and Reagents

The water utilized for experiments is ultrapure water generated by the French company Milli-Q (≥15.0 MΩ.cm). The chemical reagents used in the preparation of artificial cerebrospinal fluid are all of analytical grade (AR) and sourced from China’s Tianjin Kemio Company. All chemical reagents are stored in a light-protected and ventilated area, sealed, and samples are retrieved with tweezers for immediate use.

In this experiment, the artificial cerebrospinal fluid (ACSF) consists of 120 mM NaCl, 2.5 mM KCl, 1.25 mM NaH_2_PO_4_·2H_2_O, 26.2 mM NaHCO_3_, 1.3 mM MgSO_4_, 11 mM C_6_H_12_O_6_·H2O, and 2.5 mM CaCl_2_, with a target pH of approximately 7.3. The osmotic pressure is maintained within the range of 290–310 mOsmol/kg. Following preparation, the ACSF is cooled to a mixed state of 0 °C ice and water for later use. For the agar solution, 30 ml of distilled water is combined with 0.3 g of agar powder, mixed thoroughly, and heated to 80 °C. After stirring and cooling, the solution is allowed to stand at 40 °C to facilitate the fixation of rat brain slices before sectioning. A 10% solution of chloral hydrate is prepared by weighing 1 g of chloral hydrate, dissolving it in an appropriate amount of ultrapure water, and subsequently diluting it to 10 mL with ultrapure water. The anesthetic dose for SD rats is administered at 0.1 mL/20 g of body weight.

The solid AsHC 360 containing 2 μM arsenic was provided by the Analytical Chemistry Environmental Metallomics Laboratory at Graz University in Austria [5]. Experiments in this study utilized a solution of 15 μg As L^−1^ AsHC 360 [28]. The preparation process involved first preparing glycerol for diluting AsHC 360 (20 mL glycerol and 5 mL water). Subsequently, 25 mL of AsHC 360 solution was uniformly mixed (containing 250 μg AsHC 360, with a glycerol/water ratio of 4/1, 20 mL/5 mL) and divided into 10 centrifuge tubes of 2.5 mL each (each containing 25 μg AsHC 360). One centrifuge tube (2.5 mL) was mixed with 22.5 mL of glycerol/water (4/1) to create a 25 mL solution, which was further divided into 25 portions of 1 mL each (containing 1 μg AsHC 360). Finally, 0.75 mL of the 1 mL solution was dissolved in 50 mL of artificial cerebrospinal fluid containing oxygen (95% O_2_ + 5% CO_2_). Unless otherwise stated, all AsHC 360 assays were performed at 15 μg As L^−1^.

### 4.3. The Preparation of Hippocampal Brain Slices

The preparation of rat hippocampal brain slices followed the acute isolation method. Anesthesia was induced intraperitoneally using 10% chloral hydrate at a dose of 0.1 mL/20 g body weight [7,28,35]. Surgical instruments were pre-cooled in oxygen-enriched (95% O_2_ + 5% CO_2_) artificial cerebrospinal fluid (ACSF) during anesthesia. Once the rats were fully anesthetized, they were placed on a surgical dissecting board. A T-shaped incision was made to separate the scalp from the skull. Brain tissue was then extracted using an ophthalmic spoon and immersed in ACSF maintained at approximately 0 °C with a continuous flow of 95% O_2_ and 5% CO_2_ for several minutes to ensure adequate cooling. Subsequently, the brain tissue was removed from the ACSF and placed on an ice pillow for shaping. The hypothalamus and olfactory bulbs were removed, and the left and right hemispheres were separated. Each brain hemisphere was affixed to a sampling rod using qualitative filter paper and sealed with agar solution. Transverse slices of the rat hippocampal region, approximately 400 μm thick, were acquired using a vibratome tissue slicer (VF-2000, Compresstome, San Jose, CA, USA). Subsequently, the slices were transferred using ophthalmic forceps into a beaker containing ACSF maintained at 33 °C in an incubation chamber (BSC-PC, Harvard Apparatus, Holliston, MA, USA). Approximately 5–6 slices containing the hippocampus and suitable for experimentation were obtained from each tissue and allowed to incubate for 1 h before commencing experiments. In subsequent experiments, all brain slices were randomly assigned to control and experimental groups to minimize subjective bias.

### 4.4. Electrophysiological Experimental Setup and fEPSPs Recording Method

The electrophysiological experimental setup mainly consists of a multi-electrode array system comprising an optical microscope (model XDS-1B, OLYMPUS, Beijing, China) (Figure 1A), perfusion system (PPS2 1.3.2, Multi-Channel Systems, Reutlingen, Germany) (Figure 1B), temperature control device (TCX-Control 1.3.4, Multi-Channel Systems, Germany) (Figure 1C), MEA mainframe equipment (MEA2100 60, Reutlingen, Germany) (Figure 1D), glass electrodes (60MEA200/30iR-ITO, Reutlingen, Germany) (Figure 1F), and associated probes.

In the experiment, slices were first gently picked up with a pipette and placed onto the glass electrode. A nylon cover slip was lightly placed on top of the slice to ensure appropriate contact between the slice surface and the glass electrode array. The glass electrode array was carefully transferred to the perfusion chamber of the MEA device. Each electrode point on the glass electrode array had a diameter of 30 μm, with a distance of 200 μm between electrodes arranged in an 8 × 8 array, covering an area of 1.4 × 1.4 mm. A perfusion system was used to introduce ACSF containing AsHC 360 with 95% O_2_ + 5% CO_2_ into the perfusion chamber, with an inlet speed of 3.0 ml/min and an outlet speed of 3.3 mL/min. The inlet temperature of the system was set to 33 °C to maintain the viability of the brain slices. An inverted microscope with 10× magnification was used to adjust the position of the brain slices, and the electrode numbers corresponding to the target positions were recorded. Field excitatory postsynaptic potentials (fEPSPs) were recorded on the hippocampal Schaffer-CA1 neural pathway. Before starting the experiment, use the I/O curve function in the LTP-director software (Multi-Channel Systems) to determine the stimulation parameters that elicit the maximum amplitude of fEPSPs. Then, the recorded stimulation parameters are set to 50% of the current intensity, which maximizes the fEPSPs amplitude as the optimal stimulation. Throughout the entire fEPSPs recording period, data points were collected every minute. The software filter parameters were set to a 2-order 1 Hz high-pass filter and a 4-order 3500 Hz low-pass filter, allowing a frequency bandwidth of 1–3500 Hz.

### 4.5. Experimental Design

LTP induction involves rapid high-frequency stimulation (HFS: 100 pulses delivered at 100 Hz for 1 s) and high-frequency pulse stimulation repeated at theta rhythm frequencies (TBS: trains with an inter-train interval of 200 ms, consisting of 4 pulses at 100 Hz, one cycle lasts 230 ms, repeated 10 times for a total duration of 2300 ms). LTD induction typically requires low-frequency stimulation (LFS: Duration: 10 min, Pulse Freq: 1 Hz) [35,36]. During the induction phase of LTP, neurons are stimulated by HFS or TBS, causing synapses to transition from a resting state to an activated state, resulting in an increase in the amplitude of fEPSPs [37,38,39]. During the induction phase of LTD, neurons are stimulated by LFS, causing synapses to transition from a resting state to an activated state, resulting in a decrease in the amplitude of fEPSPs. The maintenance phase refers to the stage where LTP and LTD induction responses reach their maximum values and remain stable.

The present study investigates the impact of AsHC 360 on the bidirectional synaptic plasticity of LTP/LTD in the hippocampal Schaffer-CA1 region. The experimental protocol design is illustrated in Figure 7. Initially, we assess the effects of AsHC 360 on LTP using different induction methods (HFS, TBS). Each induction method includes a control group (without AsHC 360), a post AsHC 360 group (AsHC introduced into the system after the induction), and a pre AsHC 360 group (AsHC introduced into the system before the induction). Subsequently, we investigate the effects of AsHC 360 on LTD under the LFS induction method. This includes a control group (without AsHC 360) and two experimental groups (post AsHC 360, pre AsHC 360). Finally, we conduct experiments on Ca^2+^ concentration gradients to explore whether Ca^2+^ activity is involved in the changes in bidirectional synaptic plasticity induced by AsHC 360. The experiment on the effect of LTP consisted of two experimental groups. To explore the effect of Ca^2+^ concentration gradient on LTP-fEPSPs. AsHC 360 was added before and after the study to determine whether the Ca^2+^ activity involved in AsHC 360 caused the changes in LTP. Similarly, the experiment on the effect of LTD consists of two experimental groups. To explore the effect of Ca^2+^ concentration gradient on LTD-fEPSPs. AsHC 360 was added before and after the study to determine whether the Ca^2+^ activity involved in AsHC 360 caused changes in LTD. Each experiment has a sample size of 5.

### 4.6. Statistical Analysis

The data were normalized using LTP-analysis software and then exported in .dat raw data format. Further data processing was conducted using Origin2022 software (OriginLab, Northampton, MA, USA). Statistical analysis methods included one-way analysis of variance (ANOVA) and Tukey’s multiple comparison test. Differences were considered statistically significant when *** *p* < 0.001, ** *p* < 0.01, * *p* < 0.05, while “ns” indicated no statistical significance.

## Figures and Tables

**Figure 1 ijms-25-07223-f001:**
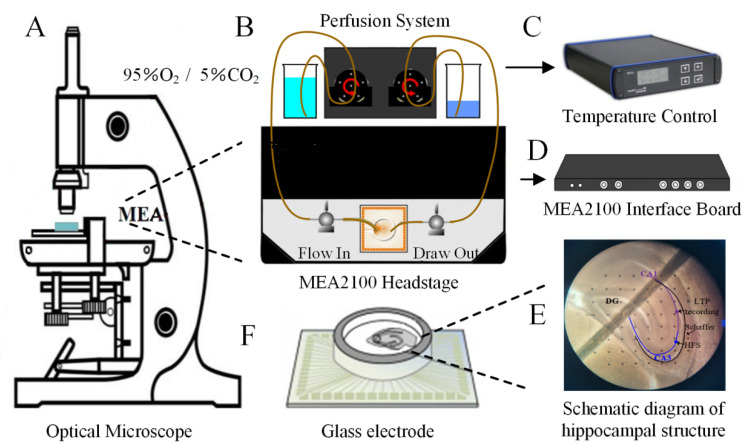
Electrophysiological Recording Process of fEPSPs (**A**): Optical Microscope (**B**): Perfusion System Setup (**C**): Temperature Control Device (**D**): MEA Interface Board (**E**): Dorsal Hippocampal Brain Slice (**F**): Glass Electrode. The MEA electrophysiology experimental platform enables observation and adjustment of the hippocampal brain region using an optical microscope. Oxygen-enriched artificial cerebrospinal fluid is perfused into the groove chamber of the glass electrode via the perfusion system to establish solution circulation. The temperature control device maintains the temperature of the artificial cerebrospinal fluid at a constant 33 °C to simulate the actual brain temperature. Stimulation and signal recording in the hippocampal brain region are controlled by the computer through the MEA system.

**Figure 2 ijms-25-07223-f002:**
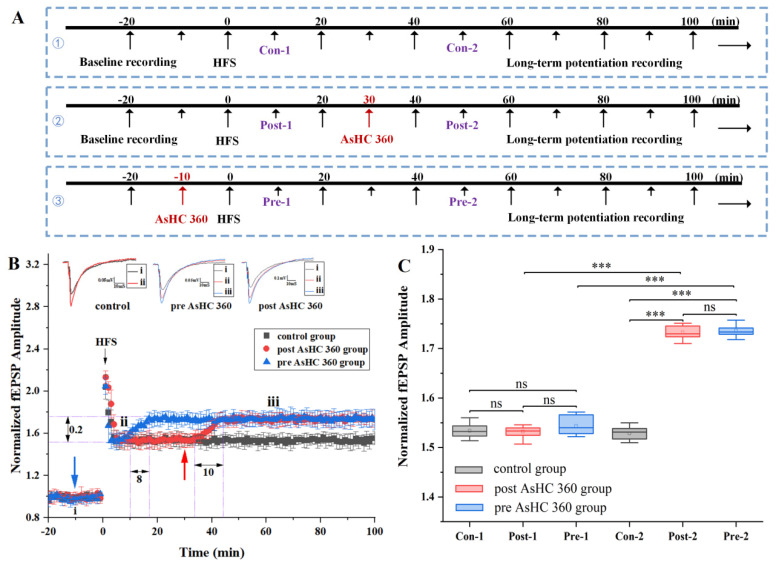
The impact of 15 μg As L^−1^ AsHC 360 on HFS-LTP (**A**): Experimental protocol ①: control Group ②: post AsHC 360 group ③: pre AsHC 360 group (**B**): Electrophysiological results of LTP-fEPSPs values for pre AsHC 360 group (blue triangle/bars), post AsHC 360 group (red circle/bars), and control group (black square/bars). Blue arrow: Introduction of AsHC 360 in the AsHC 360 group. Red arrow: Introduction of AsHC 360 in post AsHC 360 group (**C**): Boxplots and one-way ANOVA of LTP-fEPSPs values. (Con-1, Con-2; Post-1, Post-2; Pre-1, Pre-2 represent the steady-state phases corresponding to the respective experimental groups, facilitating statistical analysis within and between groups. *** denotes *p* < 0.001, ns denotes no significant difference. All groups, n = 5).

**Figure 3 ijms-25-07223-f003:**
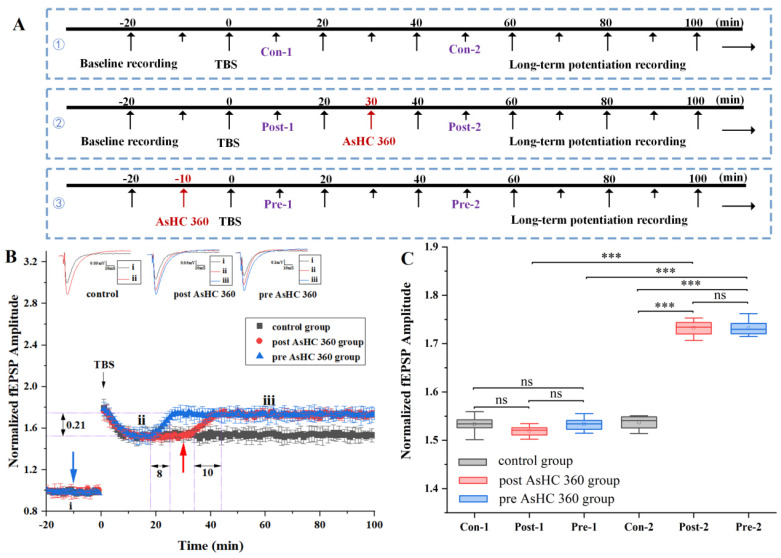
The impact of 15 μg As L^−1^ AsHC 360 on TBS-LTP (**A**): Experimental protocol (**B**): Electrophysiological results of LTP-fEPSPs values for pre AsHC 360 group (blue triangle/bars), post AsHC 360 group (red circle/bars), and control group (black square/bars). Blue arrow: Introduction of AsHC 360 in pre AsHC 360 group. Red arrow: Introduction of AsHC 360 in post AsHC 360 group (**C**): Boxplots and one-way ANOVA of the LTP-fEPSPs values. (*** represents *p* < 0.001, ns represents no significant difference. All groups, n = 5).

**Figure 4 ijms-25-07223-f004:**
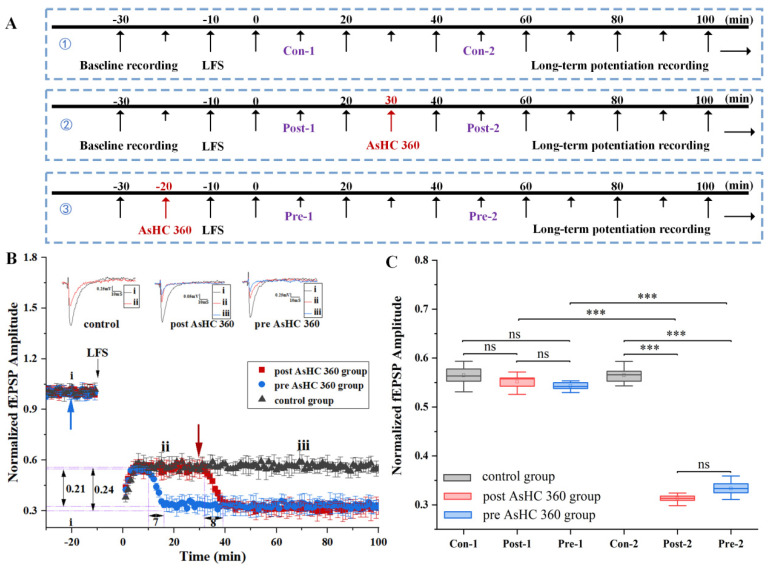
The impact of 15 μg AsL^−1^ AsHC 360 on LFS-LTD (**A**): Experimental protocol. ①: control group ②: post AsHC 360 group ③: pre AsHC 360 group (**B**): Graphical representation of LTD-fEPSPs values for pre AsHC 360 group (blue circle/bars), post AsHC 360group (red square/bars), and control group (black triangle/bars). Blue arrow: Introduction of AsHC 360 in pre AsHC 360 group. Red arrow: Introduction of AsHC 360 in post AsHC 360 group (**C**): Box plot and one-way ANOVA analysis on LTD-fEPSPs values. (*** indicates *p* < 0.001, “ns” indicates no significant difference. All groups, n = 5).

**Figure 5 ijms-25-07223-f005:**
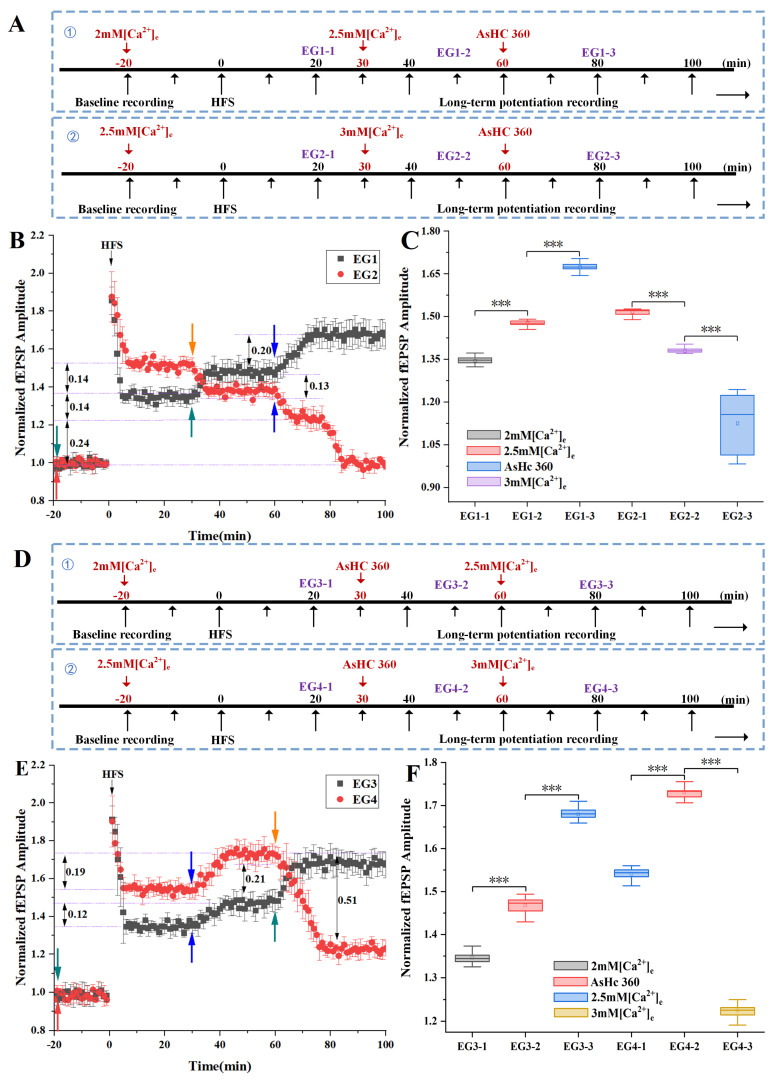
The impacts of varied Ca^2+^ Concentrations and AsHC 360 on HFS-LTP (**A**,**D**): Experimental Protocols (**B**,**E**): Electrophysiological Results. Red arrow: Introduction of (2 mM Ca^2+^) artificial cerebrospinal fluid. Cyan arrows: Introduction of (2.5 mM Ca^2+^) artificial cerebrospinal fluid. Blue arrows: Introduction of AsHC 360. Orange arrow: Introduction of (3 mM Ca^2+^) artificial cerebrospinal fluid (**C**,**F**): Box plots and one-way ANOVA analysis of LTP-fEPSPs values. (*** indicates *p* < 0.001, “ns” indicates no significant difference. All groups, n = 5).

**Figure 6 ijms-25-07223-f006:**
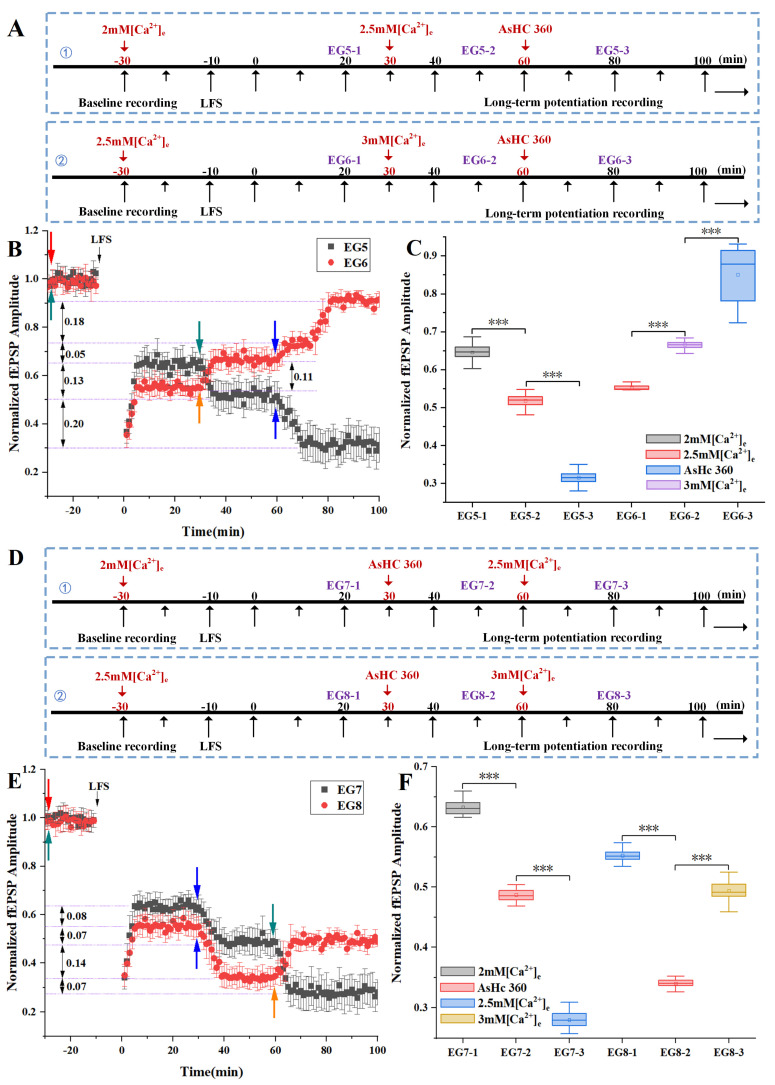
The impact of varied Ca^2+^ concentrations and AsHC 360 on LFS-LTD (**A**): Experimental protocols (**B**): Electrophysiological results (Red arrow: Introduction of 2 mM Ca^2+^ artificial cerebrospinal fluid (ACSF). Cyan arrows: Introduction of 2.5 mM Ca^2+^ ACSF. Blue arrows: Introduction of AsHC 360. Orange arrow: Introduction of 3 mM Ca^2+^ ACSF). (**C**): Box plots and one-way ANOVA analysis of LTD-fEPSPs values (**D**): Experimental protocols (**E**): Electrophysiological results (Red arrow: Introduction of 2 mM Ca^2+^ artificial cerebrospinal fluid (ACSF). Cyan arrows: Introduction of 2.5 mM Ca^2+^ ACSF. Blue arrows: Introduction of AsHC 360. Orange arrow: Introduction of 3 mM Ca^2+^ ACSF) (**F**): Box plots and one-way ANOVA analysis of LTD-fEPSPs values. (*** represents *p* < 0.001, and ns indicates no significant difference. All groups, n = 5).

**Figure 7 ijms-25-07223-f007:**
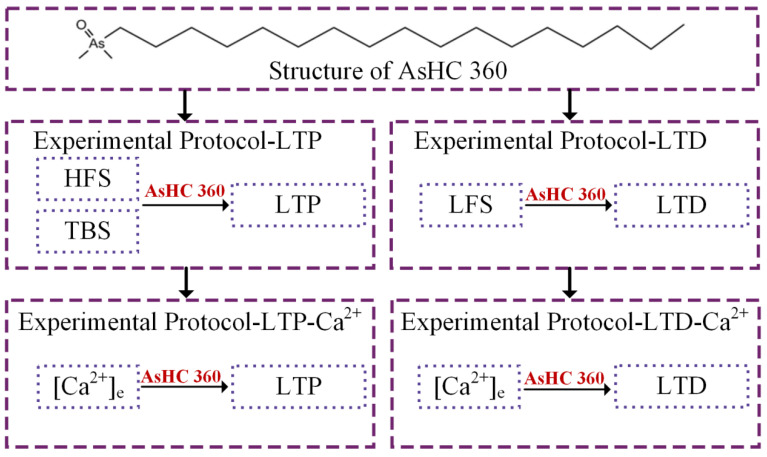
Structure of AsHC 360 and Experimental Design System Flowchart.

## Data Availability

The data supporting the findings of this study are available upon request.

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
