# Peer review of "Exploring the Effect of Arsenic-Containing Hydrocarbon on the Bidirectional Synaptic Plasticity of the Dorsal Hippocampus"

_ijms, 2024, doi:10.3390/ijms25137223_

Round 1

Reviewer 1 Report

Comments and Suggestions for Authors

In this work authors expand upon previous research conducted on AsHC 360 by evaluating its potential impact on LTD and attempt to gain some mechanistic insights on the effects of AsHC on LTP and LTD. The article is well written, but I have some questions and concerns regarding its content.

Abstract states that ’’The enhancement of bidirectional synaptic plasticity by 15 μg As L-1 AsHC 360 24 holds significant implications in improving cognitive function, treating neuro-psychiatric disorders, 25 promoting neural recovery, and enhancing brain adaptability.’’ However, this is not included in the discussion and neither is part of the conclusions of the study. In contrast, the introduction makes a point on the toxicity of AsHD. As it is now the abstract is misleading of the content and conclusions of the article.

Line 94 states that ‘’ 15 μg As L-1 AsHC 360’’ were used for the experiment, but no information is provided on how this concentration relates to physiological levels in seafood or in the milk or body after consuming it. This data is relevant as the introduction stresses the fact that AsHC has shown important toxicity. The choice seems to be mostly due to the more clear LTP effect observed in the previous article (28), but the question remains.

Line 567, for repeating the experiment, how many trains were delivered with TBS? It should be reported

Line 571, it states that ’’ LTP will increase in the amplitude of fEPSPs’’. However, it is relatively uncommon to plot amplitude of fEPSPs. It is more common to plot the slope of the rising phase of the fEPSPs. I would like to see this choice backed by the literature and explained why not use the most common and reliable slope of the fEPSPs measure.

Line 549. Please rephrase or explain: I/o function was used to find the maximum amplitude and then the stimulus for recording was set to be the one evoking 50% of the maximum amplitude, is that correct?

Line 510. Could you explain me why you use chloral hydrate to sedate your mice? To me it looks like a very particular and odd choice given its history of use in the pediatric population. See https://www.ncbi.nlm.nih.gov/pmc/articles/PMC6679948/

Line 476: I understand that some experiments might have been performed right in the day of arrival or shortly after. LTP and LTD experiments are likely affected by stress produced by traveling. Local acclimation after traveling to a new facility is highly recommended as a general best practice, and this is more important for plasticity and behavioral experiments. Please explain in more detail when, for how long and how were animals housed during and after transport prior to experiment.  

Line 467. Any particular reason for using only males?

Figure 4 a and d have errors. The second one should be EG3 and 4 I suppose

In figure 5-D should not be indicative of EG7 and EG8 conditions, or am I not understanding?

Figure 5 B. Switching from 2 to 2.5 calcium increases the effect of AsHC on LTD. However, beginning at 2.5 calcium AsHC has blocking effect on the magnitude of LTD. How is that explained?

Line 401 to 407 ‘’ AsHC 360 provides favorable conditions for membrane fusion , leading to the regulation of a series of molecular processes,..''. Here authors are providing a general mechanism of action of AsHD on synaptic plasticity that is not backed by any experiment or the literature. Cited article number 6 does not relate to synaptic plasticity.  

Author Response

The response to Reviewer 1:

Response:

We would like to extend our heartfelt appreciation for your diligent review of our manuscript and for providing us with valuable feedback and suggestions. Your insightful comments have been instrumental in improving the quality and significance of our research. We have carefully considered each of your comments and have made the necessary revisions to address the concerns raised. Your thorough evaluation has been immensely beneficial, and we believe it has contributed to enhancing the overall clarity and robustness of our work. We diligently address these concerns in our revisions, and we look forward to resubmitting an improved version of the paper for your consideration.

Q 1 Abstract states that “The enhancement of bidirectional synaptic plasticity by 15 μg As L-1 AsHC 360 24 holds significant implications in improving cognitive function, treating neuro-psychiatric disorders, 25 promoting neural recovery, and enhancing brain adaptability.” However, this is not included in the discussion and neither is part of the conclusions of the study. In contrast, the introduction makes a point on the toxicity of AsHD. As it is now the abstract is misleading of the content and conclusions of the article.  

Response:

Thank you for raising these insightful questions.

We revised the introduction and discussion sections of the article to avoid any misleading information. Additionally, in the discussion section, we explored the impact of AsHC 360 on improving cognition and treating neuropsychiatric disorders.

Line: 39-53, 496-503.

Q 2 Line 94 states that “15 μg As L-1 AsHC 360” were used for the experiment, but no information is provided on how this concentration relates to physiological levels in seafood or in the milk or body after consuming it. This data is relevant as the introduction stresses the fact that AsHC has shown important toxicity. The choice seems to be mostly due to the more clear LTP effect observed in the previous article (28), but the question remains. 

Response:

In our initial exploration of the effects of various concentrations of AsHC 360 on the maintenance phase of LTP, we found that 15 μg As L-1 enhances LTP maintenance [28]. Therefore, we continued to use the 15 μg As L-1 dosage in this study.

References:

Zheng, Y., Tian, C., Dong, L., Tian, L., Glabonjat, R. A., & Xiong, C. (2021). Effect of arsenic-containing hydrocarbon on the long-term potentiation at Schaffer Collateral-CA1 synapses from infantile male rat. Neurotoxicology, 84, 198-207.

Q 3 Line 567, for repeating the experiment, how many trains were delivered with TBS? It should be reported.

Response:

We have made the corresponding additions and revisions.

Line: 616-618.

Q 4 Line 571, it states that “LTP will increase in the amplitude of fEPSPs”. However, it is relatively uncommon to plot amplitude of fEPSPs. It is more common to plot the slope of the rising phase of the fEPSPs. I would like to see this choice backed by the literature and explained why not use the most common and reliable slope of the fEPSPs measure. 

Response:

In studies of synaptic plasticity like LTP, the amplitude of fEPSPs (field excitatory postsynaptic potentials) is a crucial metric used to assess changes in synaptic transmission efficiency, supported by numerous studies [37-39]. Similarly, measurement of fEPSPs slope is considered more common and reliable in certain situations for evaluation.

While measurements of fEPSPs slope during the rising phase offer advantages in describing dynamic changes in synaptic transmission, under specific experimental conditions, measuring the amplitude of fEPSPs can provide a more concise and direct indicator of synaptic plasticity changes. Therefore, using fEPSPs amplitude as a measurement metric is also considered reasonable and widely accepted.

References:

Kopanitsa, M. V., Afinowi, N. O., & Grant, S. G. (2006). Recording long-term potentiation of synaptic transmission by three-dimensional multi-electrode arrays. BMC neuroscience, 7, 1-19.

Chafai, M., Corbani, M., Guillon, G., & Desarménien, M. G. (2012). Vasopressin inhibits LTP in the CA2 mouse hippocampal area. PLoS One, 7(12), e49708.

Ji, Q., Yang, Y., Xiong, Y., Zhang, Y. J., Jiang, J., Zhou, L. P., ... & Zhu, Z. R. (2023). Blockade of adenosine A2A receptors reverses early spatial memory defects in the APP/PS1 mouse model of Alzheimer’s disease by promoting synaptic plasticity of adult-born granule cells. Alzheimer's Research & Therapy, 15(1), 187.

Q 5 Line 549. Please rephrase or explain: I/o function was used to find the maximum amplitude and then the stimulus for recording was set to be the one evoking 50% of the maximum amplitude, is that correct? 

Response:

It's like this. We have rephrased it in the article.

Line: 596-599.

Q 6 Line 510. Could you explain me why you use chloral hydrate to sedate your mice? To me it looks like a very particular and odd choice given its history of use in the pediatric population. See https://www.ncbi.nlm.nih.gov/pmc/articles/PMC6679948/ 

Response:

Using chloral hydrate to sedate mice is indeed a relatively specific choice. Although there is literature indicating potential toxicity of chloral hydrate to the liver and kidneys, with side effects including respiratory depression, hypotension, and cardiac arrhythmias, it remains a traditional sedative and anesthetic agent that effectively and rapidly sedates experimental animals, making them suitable for various procedures. Moreover, previous work on AsHC 360 has used chloral hydrate, facilitating comparability. In future studies, we also plan to explore the use of anesthetic agents such as isoflurane to sedate mice, to further observe experimental differences.

References:

Dong, L., Zhao, L., Tian, L., Zhao, W., Xiong, C., & Zheng, Y. (2023). AsHC 360 Exposure Influence on Epileptiform Discharges in Hippocampus of Infantile Male Rats In Vitro. International Journal of Molecular Sciences, 24(23), 16806.

Zheng, Y., Tian, C., Dong, L., Tian, L., Glabonjat, R. A., & Xiong, C. (2021). Effect of arsenic-containing hydrocarbon on the long-term potentiation at Schaffer Collateral-CA1 synapses from infantile male rat. Neurotoxicology, 84, 198-207.

Zheng, Y., Ma, X. X., Dong, L., Ma, W., & Cheng, J. H. (2019). Effects of uninterrupted sinusoidal LF-EMF stimulation on LTP induced by different combinations of TBS/HFS at the Schaffer collateral-CA1 of synapses. Brain Research, 1725, 146487.

Q 7 Line 476: I understand that some experiments might have been performed right in the day of arrival or shortly after. LTP and LTD experiments are likely affected by stress produced by traveling. Local acclimation after traveling to a new facility is highly recommended as a general best practice, and this is more important for plasticity and behavioral experiments. Please explain in more detail when, for how long and how were animals housed during and after transport prior to experiment.  

Response:

We have made the corresponding additions and revisions.

Line: 506-523.

Q 8 Line 467. Any particular reason for using only males?

Response:

1: The stability of hormone levels varies, with male mice having relatively stable hormone levels, whereas female mice experience hormonal fluctuations with their estrous cycle. These fluctuations can potentially affect the stability and reproducibility of experimental results.

2: Standardization of experiments is crucial, and many neuroscience and behavioral experiments traditionally use male mice. This practice facilitates easier comparison and standardization of research findings with existing literature.

3: Simplification of experimental design is important in the early stages of research. Using animals of a single sex can streamline both the experimental setup and data analysis. Once preliminary results are obtained, the study can be expanded to include female animals to verify the generalizability of the findings.

Q 9 Figure 4 a and d have errors. The second one should be EG3 and 4 I suppose

Response:

The text has been corrected. In Figure 4D, EG1-1, EG1-2, EG1-3; EG2-1, EG2-2, EG2-3 should be labeled as EG3-1, EG3-2, EG3-3; EG4-1, EG4-2, EG4-3.

Q 10 In figure 5-D should not be indicative of EG7 and EG8 conditions, or am I not understanding? 

Response:

You are correct in your understanding, and we have made the necessary corrections in the text.

Q 11 Figure 5 B. Switching from 2 to 2.5 calcium increases the effect of AsHC on LTD. However, beginning at 2.5 calcium AsHC has blocking effect on the magnitude of LTD. How is that explained? 

Response:

In Figure 5B, the experimental group EG5 corresponds to the experimental protocol shown in Figure 5A-①. The 2 mM [Ca2+]e phase spans from -30 minutes to 30 minutes, and the 2.5 mM [Ca2+]e phase spans from 30 minutes to 60 minutes. During the transition from 2 mM [Ca2+]e to 2.5 mM [Ca2+]e, the brain slices are exposed to ACSF with different calcium ion concentrations, without the introduction of AsHC 360 at this stage. As the calcium ion concentration increases, it promotes the occurrence of LTD, resulting in a decrease in fEPSPs amplitude.At t=60 minutes, AsHC 360 is introduced. By further affecting the changes in calcium ion levels, AsHC 360 further promotes the occurrence of LTD, causing the fEPSPs amplitude to decrease even more.

Q 12 Line 401 to 407 “ AsHC 360 provides favorable conditions for membrane fusion , leading to the regulation of a series of molecular processes,..''. Here authors are providing a general mechanism of action of AsHD on synaptic plasticity that is not backed by any experiment or the literature. Cited article number 6 does not relate to synaptic plasticity.

Response:

Related literature has been added to the main text to support the findings. AsHC 360, as a type of arsenolipid, may provide favorable conditions for membrane fusion in cognitive processes, enhance synaptic transmission, and promote bidirectional synaptic plasticity.

Line: 430-433.

We hope that all these changes fulfil the requirements to make the manuscript acceptable for publication in International Journal of Molecular Sciences. If there are any other problems with our paper, please feel free to contact me.

Thank you again for your efforts and time for our manuscript.

Kind regards,

Yu Zheng

Reviewer 2 Report

Comments and Suggestions for Authors

General remarks

The authors used rat hippocampal slices to detect the influence of an organic arsenic compound on long term potentiation (LTP) and long term depression (LTD).

They found that the compound enhances the maintenance phase of both LTP and LTD . Moreover, they added calcium in different concentrations to the culture and found that the compound increases the effect of the calcium. 

The study design and the findings are very interesting. However, the manuscript is very badly written: in many parts of the manuscript, the introductory sentences are hidden in the last paragraphs, making the previous paragraphs very difficult to understand. I suggest a complete reorganisation of the manuscript, with solid introductions at the beginning of every chapter (I have listed a few examples below).

My other major point concerns the findings and their interpretation: the first part of the experiments, which demonstrate that the arsenic compound increases the hold phase of both LTP and LTD are convincing and correctly interpreted. But the authors claim that the seond part of the experiments, done at different calcium levels, proves that the compounds works by increasing calcium entry. I think this overinterprets the data. I do see that the compound exacerbates the effects of the different calcium levels, but that is no proof of a way of action. The authors should either explain exactly why they come to this conclusion or choose correct wording.

Detailed remarks:

intro

Line 43 – definition of ASHC 360 necessary.

Line 92 – 93, please state briefly how you conducted the experiments for the current study (that you worked with rat hippocampal slices and how).

Results

Line 104- just like the introduction, the results assume that the reader knows the experimental setup. The reader does not know about the setup so please explain it here.

lines 259-368

confusing since all of a sudden, the figure numbers are used as headlines- please reorganise.

Discussion

412-413 sentence out of place: Calcium homeostasis has reached a new equilibrium, please explain.

Lines 408-429 – are surely not new but demonstrate that the authors’ experiments worked. Please shorten but leave enough information so that the reader understands the paragraph on the influence of the AsHC360 on the calcium influx.

 Line 431 -436- are introductory and should be moved to the beginning of the chapter discussing calcium ions.

Legends

Figure 1

What do the blue labelled arrows “con-1, Post-1, Pre-1 and 2 etc” mean, what happens at these time points?

Figure 5

No legend

Author Response

The response to Reviewer 2:

Response:

Thank you for your careful review and feedback on our manuscript. We sincerely appreciate your insightful comments and suggestions. In response to your feedback, we have diligently made comprehensive revisions to the manuscript. We have addressed each of the aspects you highlighted, ensuring the manuscript is now enhanced in its quality and rigor. We believe that these changes will make the work more robust and clear for our readers. Once again, thank you for your constructive feedback, and we hope our revisions meet your expectations.

Q 1 My other major point concerns the findings and their interpretation: the first part of the experiments, which demonstrate that the arsenic compound increases the hold phase of both LTP and LTD are convincing and correctly interpreted. But the authors claim that the seond part of the experiments, done at different calcium levels, proves that the compounds works by increasing calcium entry. I think this overinterprets the data. I do see that the compound exacerbates the effects of the different calcium levels, but that is no proof of a way of action. The authors should either explain exactly why they come to this conclusion or choose correct wording.

Response:

We have revised the issues you raised in the text and selected the correct expressions.

Line: 21-23.

Q 2 Line 43 – definition of ASHC 360 necessary. 

Response:

The text has been supplemented with additional information about AsHC 360.

Line: 41.

Q 3 Line 92 – 93, please state briefly how you conducted the experiments for the current study (that you worked with rat hippocampal slices and how).

Response:

This section has been supplemented with an introduction to AsHC 360, and a detailed description has been provided in the later chapters.

Line: 93-96, 555-574.

Q 4 Line 104- just like the introduction, the results assume that the reader knows the experimental setup. The reader does not know about the setup so please explain it here.

Response:

The section has been extensively revised in the text.

Line: 107-122.

Q 5 lines 259-368confusing since all of a sudden, the figure numbers are used as headlines- please reorganise.

     Response:

The section has been thoroughly revised in the manuscript.

Line: 250-253, 257-260, 280-283, 291-294, 324-327, 332-335, 357-360, 369-371.

Q 6 412-413 sentence out of place: Calcium homeostasis has reached a new equilibrium, please explain.

Response:

The expression of this sentence was not accurate. Here, we intended to convey that all phenomena were analyzed under steady-state conditions. We have already removed this sentence from the manuscript.

Q 7 Lines 408-429 – are surely not new but demonstrate that the authors’ experiments worked. Please shorten but leave enough information so that the reader understands the paragraph on the influence of the AsHC360 on the calcium influx.

Response:

The text has already been revised in this part.

Line: 439-460.

Q 8 Line 431 -436- are introductory and should be moved to the beginning of the chapter discussing calcium ions. 

Response:

The text has already been revised in this section and moved to the beginning of the calcium ion discussion chapter.

Line: 439-442.

Q 9 Figure 1. What do the blue labelled arrows “con-1, Post-1, Pre-1 and 2 etc” mean, what happens at these time points?

Response:

Con-1 and Con-2 represent the control group; Post-1 and Post-2 represent the experimental group (AsHC 360 added after LTP induction); Pre-1 and Pre-2 represent the experimental group (AsHC 360 added during the baseline stage before LTP induction). This division facilitates statistical analysis both within and between groups. The explanation for this division has already been included in the text.

These stages represent the steady state of the corresponding experimental phases. For example, Post-1 represents the stable phase of the experimental group after LTP induction, but before the addition of AsHC 360. Post-2 represents the stable phase of the experimental group after AsHC 360 has been added on top of the established LTP. This division facilitates statistical analysis within the experimental groups.

Line: 160-162.

Q 10 Figure 5 no legend

Response:

Figure 5 includes a legend and has already been supplemented and revised.

Line: 395-404.

We hope that all these changes fulfil the requirements to make the manuscript acceptable for publication in International Journal of Molecular Sciences. If there are any other problems with our paper, please feel free to contact me.

Thank you again for your efforts and time for our manuscript.

Kind regards,

Yu Zheng

Round 2

Reviewer 2 Report

Comments and Suggestions for Authors

In my opinion the manuscript has been improved and is almost ready for publication.  I still have one major concern that I would like addressed, though: I see no proof in the data that the compound has an influence on calcium levels. So please change the wording throughout, clarifying that this is your assumption rather than a proof.

Author Response

The response to Reviewer :

Response:

Thank you for your careful review and feedback on our manuscript. We sincerely appreciate your insightful comments and suggestions. In response to your feedback, we have diligently made comprehensive revisions to the manuscript. We have addressed each of the aspects you highlighted, ensuring the manuscript is now enhanced in its quality and rigor. We believe that these changes will make the work more robust and clear for our readers. Once again, thank you for your constructive feedback, and we hope our revisions meet your expectations.

Q 1 In my opinion the manuscript has been improved and is almost ready for publication. I still have one major concern that I would like addressed, though: I see no proof in the data that the compound has an influence on calcium levels. So please change the wording throughout, clarifying that this is your assumption rather than a proof.

Response:

We have adopted the reviewers' suggestions and revised the wording of the article.

Line: 22-23, 413-415, 466-467.

We hope that all these changes fulfil the requirements to make the manuscript acceptable for publication in International Journal of Molecular Sciences. If there are any other problems with our paper, please feel free to contact me.

Thank you again for your efforts and time for our manuscript.

Kind regards,

Yu Zheng